# Higher-level spatial prediction in natural vision across mouse visual cortex

**Micha Heilbron**[1,2]*, **Floris P. de Lange**[1]

1 Donders Institute for Brain, Cognition and Behaviour, Radboud University, Nijmegen, Netherlands,
2 Amsterdam Brain and Cognition, University of Amsterdam, Amsterdam, Netherlands

* m.heilbron@uva.nl

## Abstract

Theories of predictive processing propose that sensory systems constantly predict incoming signals, based on spatial and temporal context. However, evidence for prediction in sensory cortex largely comes from artificial experiments using simple, highly predictable stimuli, that arguably encourage prediction. Here, we test for sensory prediction during natural scene perception. Specifically, we use deep generative modelling to quantify the spatial predictability of receptive field (RF) patches in natural images, and compared those predictability estimates to brain responses in the mouse visual cortex—while rigorously accounting for established tuning to a rich set of low-level image features and their local statistical context—in a large scale survey of high-density recordings from the Allen Institute Brain Observatory. This revealed four insights. First, cortical responses across the mouse visual system are shaped by sensory predictability, with more predictable image patches evoking weaker responses. Secondly, visual cortical neurons are primarily sensitive to the predictability of higher-level image features, even in neurons in the primary visual areas that are preferentially tuned to low-level visual features. Third, unpredictability sensitivity is stronger in the superficial layers of primary visual cortex, in line with predictive coding models. Finally, these spatial prediction effects are independent of recent experience, suggesting that they rely on long-term priors about the structure of the visual world. Together, these results suggest visual cortex might predominantly predict sensory information at higher levels of abstraction—a pattern bearing striking similarities to recent, successful techniques from artificial intelligence for predictive self-supervised learning.

**Data availability statement:** The Allen Institute Visual Coding Dataset is freely available online, via the Allen Institute: https: //observatory.brain-map.org/visualcoding/. All code and files needed to reproduce all results and analyses are published on the Donders Data Repository at the following permalink: https://doi.org/10.34973/53ew-h870.

**Funding:** FPdL was supported by Netherlands Organisation for Scientific Research (NWO), Vici grant (VI.C.231.043 to F.P.d.L.). MH and FPdL were supported by the European Research Council (ERC) under the European Union's Horizon 2020 research and innovation programme (grant agreement No. 101000942, SURPRISE, to FPdL). The funders had no role in study design, data collection and analysis, decision to publish, or preparation of the manuscript.

**Competing interests:** The authors have declared that no competing interests exist.

## Author summary

How does the brain make sense of the constant stream of visual information? A popular theory suggests the brain is not a passive receiver but an active predictor, constantly generating predictions about incoming sensory input. We tested this idea by analysing neural responses of thousands of brain cells of mice watching natural images. Using an AI model, we could quantify how predictable any specific image patch was given its surroundings. As predicted by the theory, we found that the brain cells indeed responded less to more predictable parts of an image. This effect appears based on long-term knowledge, as it was independent of the animal's recent experience with the images. Strikingly, we discovered that even in the earliest stages of visual processing, the brain is most sensitive to the predictability of complex patterns and textures, not that of simple features like edges. This strategy of predicting specifically the complex (high-level) information mirrors recent breakthroughs in AI, suggesting that both brains and these recent AI systems may learn to understand the visual world through a similar, predictive process.

## 1 Introduction

Theories of *predictive processing* propose that neural information processing critically relies on comparing (bottom-up) incoming signals to self-generated (top-down or locally recurrent) predictions, derived from a generative model [7–9], in order to drive both sensory inference and self-supervised learning.

Over the past decade, predictive processing has become very influential, as a body of findings accumulated that support the framework [9–11]. Most of this work shows that stimuli evoke weaker responses when they are predictable and stronger responses when they are surprising; a phenomenon known as *expectation suppression* which is typically interpreted as reflecting sensory prediction errors. In perception, such effects are typically studied by experimentally *manipulating* stimulus probability—for instance by extensively exposing observers to one stimulus predictably following another stimulus (e.g. [12–14]; see [10,15] for review). One limitation of this line of work is that it is unclear whether effects found under such artificial (and prediction-encouraging) conditions generalise to natural conditions. Typically, there are only a handful of stimuli that can occur in these experiments (e.g., a clockwise or counter-clockwise grating), whereas the realm of possibilities in natural conditions is nearly limitless. Moreover, because most studies use relatively coarse comparisons (e.g. expected vs unexpected stimulus), it typically remains unclear at what level of granularity sensory predictions are made: does sensory cortex make predictions about low-level, mid-level or high-level stimulus category features, or perhaps all of these at the same time?

Advances in generative AI are allowing for a different way of studying predictive processing that can address both these issues. Instead of experimentally *manipulating* stimulus predictability, this involves *estimating* stimulus predictability using generative models, to test if predictable stimuli (along a certain dimension) evoke an

attenuated neural response. Critically, this enables (i) studying predictions in natural conditions (i.e. without experimentally imposing the prediction on the observer), and (ii) assessing their granularity, as predictability can be computed at multiple levels of abstraction [16,17]. Recently, [3] introduced a framework to apply this approach to spatial predictability in visual scene perception. They recorded multi-unit activity from primary visual cortex in macaques viewing natural images. Then, using Deep Neural Networks (DNNs) trained for in-painting, they quantified the spatial predictability of image patches that were centered on the classical receptive field of the neurons, and found that less predictable patches evoked higher firing rates in macaque V1. Strikingly, in an additional analysis focusing specifically on a later phase of processing (200-600 ms after stimulus onset) they found the unpredictability of high-level features of the image part drove activity most strongly. However, their analysis was based on a single area in macaque visual cortex, leaving it unclear how their findings would extend beyond primate vision and across the visual cortical hierarchy.

Here, we build on their approach, and apply it to the Allen Institute Neuropixels dataset [1]. This is a high-quality, comprehensive dataset of thousands of individual neurons across multiple cortical layers of the entire mouse visual cortical system of 32 mice. This dataset covers an extensive stimulus battery that includes controlled stimuli to characterise the tuning properties of individual neurons. This allowed us to quantify the spatial predictability for every stimulus, for every receptive field of every individual unit across the visual cortical hierarchy, while controlling for receptive field visual features and local statistical context (including contrast energy, orientation content, orientation consistency and coherence). This way, we ask four questions: (i) are cortical responses to natural images in visual cortex modulated by spatial predictability?; (ii) at what level of granularity do neurons in the visual cortex predict, and how does this differ across different areas of the visual system?; (iii) are prediction effects stronger in superficial compared to deep layers of cortex?; (iv) are these effects based on short-term online-learnt expectations, or on more long-term structural priors, independent of recent experience?

To preview, we find that spatial predictability modulates cortical responses across the mouse visual cortex, with neurons most sensitive to high-level predictability even in primary visual areas. These spatial predictability effects were independent of recent experience, and in V1 they were strongest in superficial layers. Together, these results shed new light on sensory prediction, with implications for models of predictive processing and self-supervised learning in sensory cortex.

## 2 Results

We analyzed the Allen Institute Neuropixels dataset [1], focusing on spiking responses to the "natural scenes" stimulus set, in which head-fixed, awake mice were monocularly presented with full-field natural images, for 250 ms per presentation and 50 repetitions per stimulus (Fig 1a). Critically, stimuli were presented in a pseudo-random order without predictable temporal structure or prediction-encouraging task, allowing us to test for prediction in complex visual processing, without extraneous highly predictable structure. For our analyses, we selected neurons with well-defined receptive fields (see Methods for selection criteria) across six cortical visual areas that exhibit a hierarchical organization as established by [1]. While our analyses encompass all six areas, our main focus was on primary visual cortex (V1), as it offered substantially greater coverage with the highest number of included neurons across the sample of mice.

### 2.1 Firing rates throughout visual cortex are modulated by spatial predictability

We first asked whether neural responses were modulated by overall spatial predictability. To compute spatial predictability for every patch in every image, we masked out the image patch that spans the receptive field of an image, then use a deep generative model to predict or "fill-in" the missing image patch, and compare the predicted image patch with the actual image patch to compute overall predictability. While this method is designed to quantify predictability in complex natural scenes, it can also recapitulate classic contextual effects with simple parametric stimuli (see S1 and S2 Figs).

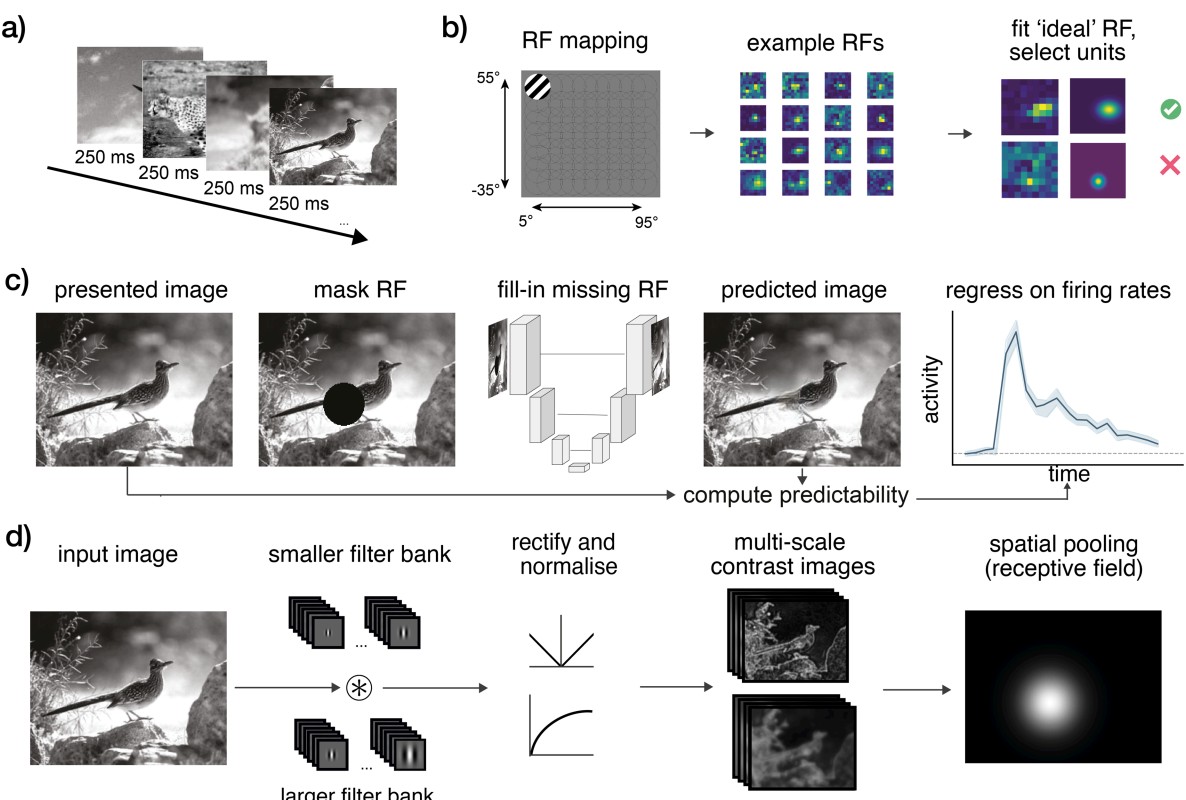

**Fig 1**. **A) Dataset.** We analysed cortical responses to natural images from the Allen Institute Visual Coding Neuropixels dataset, in which mice viewed full-field natural images for 250 ms per image, for a total of 5,900 trials [1]. **B)** Prior to the natural scenes, flashing gratings were presented allowing to map the receptive field (RF) of every unit; we selected neurons based on their RF characteristics (see *Methods*). **C)** Spatial predictability analysis: for every image and every neuron, the area spanning the receptive field was masked out and filled-in using a reconstructive autoencoder with skip connections (UNet; [2]). The predicted image patch was compared to the actual patch to compute the spatial predictability of that patch [3]. **D)** To control for low-level image statistics, we generated multi-scale first-order (contrast energy) and second-order (spatial coherence) contrast images using Gabor filter banks ([4,5]). From these, we derived a comprehensive set of baseline statistics (including measures of local contrast, orientation content and homogeneity, and coherence; see *Methods*). In all panels, example images are from the Allen Institute Visual Coding stimuli, and originally from the van Hateren natural image database [6].

Critically, this predictability calculation was performed for every classical receptive field of every unit of every image individually (see Fig 1c and *Methods*). Cortical responses were assessed as firing rate activity time-courses (expressed as fold-change with respect to baseline) for every neuron, for every stimulus.

To isolate the influence of image patch unpredictability from known drivers of visual cortical activity, we first fitted a robust baseline regression model to the firing rates at every timepoint. This baseline model incorporated a series of non-prediction-related variables, including running speed and a comprehensive set of low-level image statistics (e.g., local contrast energy across spatial frequencies, orientation content, and circular variance, indexing contour homogeneity; detailed in *Methods* 4.6). To evaluate whether unpredictability significantly modulated firing rate, we then evaluated whether adding our image patch unpredictability metric could explain unique, additional variance beyond this comprehensive baseline. To test this, we computed $\Delta R$—the increase in cross-validated prediction performance when unpredictability was added to the baseline model (Fig 2a, 2b). Note that this cross-validated metric provides a conservative test for the presence of an effect, rather than a direct estimate of the effect size magnitude.

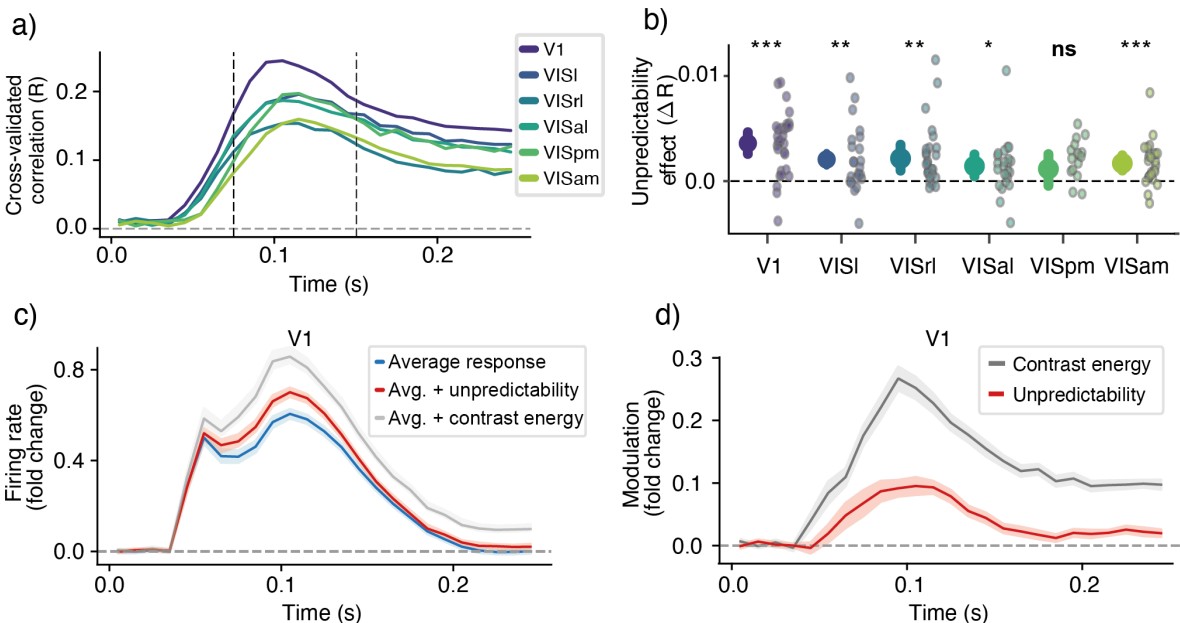

**Fig 2**. **a) Cross-validated prediction performance of the baseline model across the visual cortical areas, recapitulating the hierarchical organization established in [1]. b)** Average increase in cross-validated performance between 75-150 ms—see dotted lines in panel a)—when adding unpredictability to the baseline model. Small dots indicate individual animals; large dots with error bar indicate mean across animals and bootstrapped 95% CI. Note, $\Delta R$ provides a conservative test for the presence of an effect; for effect size, see $\beta$ coefficients in panels c-d. **c)** Effect of unpredictability and contrast on average response in V1. Curves show the average response (blue curve), and the average response plus $\beta$ coefficients of unpredictability (red) and contrast (gray). Note that firing rate is expressed as fold-change with respect to baseline (0-20 ms; see *Methods*). **d)** Same as panel c) but showing just the modulation function ($\beta$ coefficients), rather than effect on average response. In both panels, curves show average coefficient across neurons for each animal, averaged across animals; shading shows bootstrapped 95% CI across animals. Note that in all regressions, predictor features are z-scored, making coefficients reflect standardised modulation strength.

This revealed robust effects of unpredictability, both in V1 (mean $\Delta R = 0.0036$, 95% CI [0.0026,0.0046], bootstrap *t*-test with Bonferroni correction: $p < 0.0001$), and across downstream visual areas (VISl: mean $\Delta R = 0.0021$, 95% CI [$8.45 \times 10^{-4}$, 0.0034], $p = 0.006$; VISrl: mean $\Delta R = 0.0022$, 95% CI [0.0010,0.0034], $p = 0.0012$; VISal: mean $\Delta R = 0.0014$, 95% CI [$4.47 \times 10^{-4}$, 0.0026], $p = 0.0144$; VISam: mean $\Delta R = 0.0017$, 95% CI [$9.77 \times 10^{-4}$, 0.0025], $p < 0.0001$), with the exception of VISpm where the effect was not significant (mean $\Delta R = 0.0012$, 95% CI [$-4.52 \times 10^{-4}$, 0.0025], $p = 0.282$). Together, this shows that cortical responses are indeed sensitive to spatial unpredictability, across multiple cortical regions.

Having established that spatial unpredictability impacts the neural response, we next examined the coefficients of the regression model to characterize the nature of this effect. The coefficients of this regression model can be interpreted as 'modulation functions' that quantify how a variable modulates the firing rate over time. As can be seen from Fig 2c, 2d, we observed a clear positive modulation by unpredictability (average coefficient between 75–150 ms: $M = 0.077$, 95% CI [0.059,0.096], $p < 0.0001$, bootstrap t-test across animals). This indicates that more predictable image patches are associated with weaker neural responses (expectation suppression). For reference, the effect of unpredictability can be compared to the effect of local contrast energy in a unit's preferred contrast channel. This reveals that the effect of contrast is stronger than the effect of unpredictability (paired difference between 75–150 ms: $M = 0.132$, 95% CI [0.108,0.157], $p < 0.0001$). Critically, the contrast effect appears to modulate the neural response earlier (Fig 2c, 2d; average latency to 50% peak: $M = 71.0$ ms, SE = [68.8,73.2] vs. unpredictability: $M = 77.3$ ms, SE = [75.1,79.6]), in line with the notion that predictability effects rely more on recurrent processing.

## 2.2 Opposite tuning for stimulus features and feature predictability

After confirming that overall spatial predictability modulated responses across mouse visual cortex, we next assessed the representational content of these predictions. This was done by estimating the neural effect not just of overall predictability, but of predictability at multiple levels of abstraction [3,16]. These multi-level predictability estimates were obtained by comparing the actual and predicted patch at increasingly abstract feature spaces. For this we used the convolutional layers of AlexNet [18], a shallow CNN with a good fit to mouse visual cortex [19]. Because early CNN layers extract simple features (e.g. edges) and higher levels more complex ones (e.g. textures), performing the comparison at each layer allows to quantify predictability at multiple levels of abstraction (Fig 3a, 3b).

Focusing on V1 first, we computed the unpredictability sensitivity as the average coefficient between 75-150 ms, for each level of unpredictability. This revealed a clear positive relationship (see Fig 3c): firing rates in V1 were least sensitive to low-level unpredictability, and most sensitive to high-level unpredictability; and this pattern was highly consistent across animals (mean slope = 0.149, 95% CI [0.096, 0.210], bootstrap: $p < 0.0001$), suggesting that early visual cortex predicts higher rather than lower-level visual features. While this is in line with recent observations ([3,20]) the pattern is striking

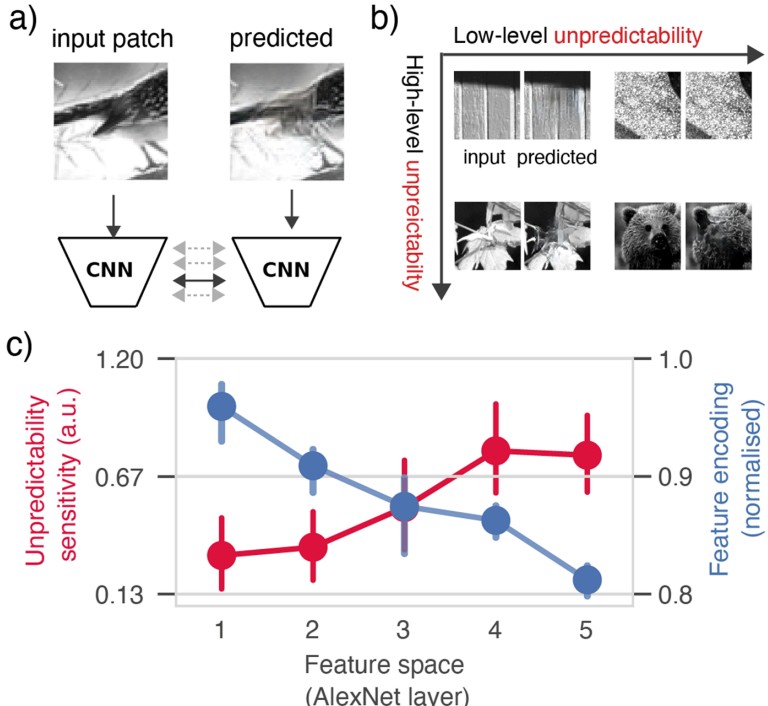

**Fig 3**. **a) Multi-level predictability analysis.** To compute unpredictability at multiple levels, the predicted RF-patch and actual input RF-patch were fed into a CNN to extract a series of increasingly abstract features. The predicted and actual input patch were then compared for each level separately, to separately quantify the predictability of low- and high-level features. **b)** Dissociation between high-level and low-level predictability. Some illustrative example image patches (of stimuli from [1,6]) at four quadrants of the high-level vs low-level unpredictability axes. One common example are textures (grass, moss, sandstone), for which the exact composition of low-level features (edges and orientations) are highly unpredictable, but the higher-level structure (the texture) is perfectly predictable; hence right top corner. See *Methods* for more details, and S3 Fig for more examples. **c)** Selectivity for visual features (blue) and visual feature predictability (red) in V1. Image feature encoding (blue) shows normalised encoding, i.e. correlation coefficient divided by the maximum encoding performance per unit, before averaging across units. Unpredictability sensitivity is defined as average coefficient in the 75-150 ms timewindow (see Fig 2). Dots show mean across animal-level averages; error bars show bootstrapped 95% confidence intervals. Importantly positive trend for unpredictability sensitivity was absent in a control analysis using synthetic noise images, confirming it reflects neural tuning (see S7 Fig).

since the *unpredictability-sensitivity* to higher versus lower-level features is the opposite of the well-characterised *feature-sensitivity* of V1. Indeed, when we examined the tuning profile of the neurons, by performing a more standard encoding-modelling based feature-sensitivity analysis—i.e. predicting firing rates as a function of CNN image features, rather than the unpredictability—we found that V1 was tuned to low-level, instead of high-level, features. Namely, V1 firing rates were best predicted by low-level image features (from early CNN layers), and progressively worse by higher-level image features. This negative relationship recapitulates traditional bottom-up feature tuning, and was again highly consistent across animals (mean slope = –0.034, 95% CI [–0.039, –0.027], bootstrap test across animals: $p < 0.0001$).

This dissociation between feature-sensitivity (preferring low-level features) and unpredictability-sensitivity (preferring high-level features) provides clear evidence that the high-level unpredictability effects are distinct from V1's basic tuning to simple image features (see Discussion). Interestingly, when we repeated the same analysis for other downstream areas in the mouse visual cortex, we found largely the same pattern of effects: feature sensitivity was highest low-level image features, but unpredictability sensitivity was highest for high-level features—although we also found that, for encoding, later level visual areas are more sensitive to higher-level features than early visual areas (see S4 and S5 Figs). The relatively modest differences in selectivity between V1 and higher visual areas is in line with earlier encoding analyses of mouse visual cortex [19].

As a final confirmatory control, we repeated the predictability-sensitivity analysis, but now using cross-validation: quantifying predictability effects as the increase in regression model fit when adding each unpredictability estimate to the baseline model. This reveals a very similar positive relationship for V1 (S6a Fig), that is highly consistent across animals (mean slope across $\Delta R$ = 7.56 $\times$ 10$^{-4}$, 95% CI [4.68 $\times$ 10$^{-4}$, 1.09 $\times$ 10$^{-3}$], bootstrap $t$-test: $p < 0.0001$). Indeed, we find again a similar pattern across all cortical areas, where unpredictability effects are strongest for higher-level unpredictability and weakest for low-level unpredictability (S6b Fig).

Finally, we explored whether the timing of these effects also varied across the different levels of unpredictability. However, a dedicated latency analysis did not reveal any clear or consistent patterns (see S9 Fig). This null-result should be interpreted with caution, as the analysis may not have been sensitive enough to detect more subtle latency differences between the conditions.

Together, we observed that predictability sensitivity does not simply follow the neuron's bottom-up feature sensitivity. Instead, we found a distinct preference for unpredictability of higher-level features. This results in a highly consistent pattern of opposite tuning: neurons are most sensitive to lower-level image features, but higher-level feature predictability.

## 2.3 Predictability effects are strongest in superficial layers of primary visual cortex

A distinguishing feature of predictive coding models is that they postulate two distinct populations of neurons: error neurons that signal the mismatch between predictions and sensory input, and prediction or representation neurons that encode internal models of the environment [8–10,21]. In hierarchical predictive coding models, predictions flow downward through the cortical hierarchy while prediction errors propagate forward. Because in cortex, feedforward signals originate from superficial layers (2/3) while feedback signals stem from deep layers (5/6), this anatomical asymmetry maps directly onto the functional asymmetry in predictive coding. Namely, superficial-layer neurons should primarily signal prediction errors, whereas deep-layer neurons should encode predictive representations of sensory input.

The Visual Coding dataset allowed to test this hypothesized laminar organization, as it provides simultaneous recordings across all cortical layers of the visual system (Fig 4a). We took this opportunity by classifying neurons as either superficial (layer 2/3) or deep (layer 5/6) based on their precise recording depths (see Methods for detailed classification criteria). Then, we tested whether predictability effects (assuming they reflect sensory prediction errors) were indeed strongest in superficial layers of cortex. To this end, we quantified the modulation of firing rates (75-150 ms) by overall

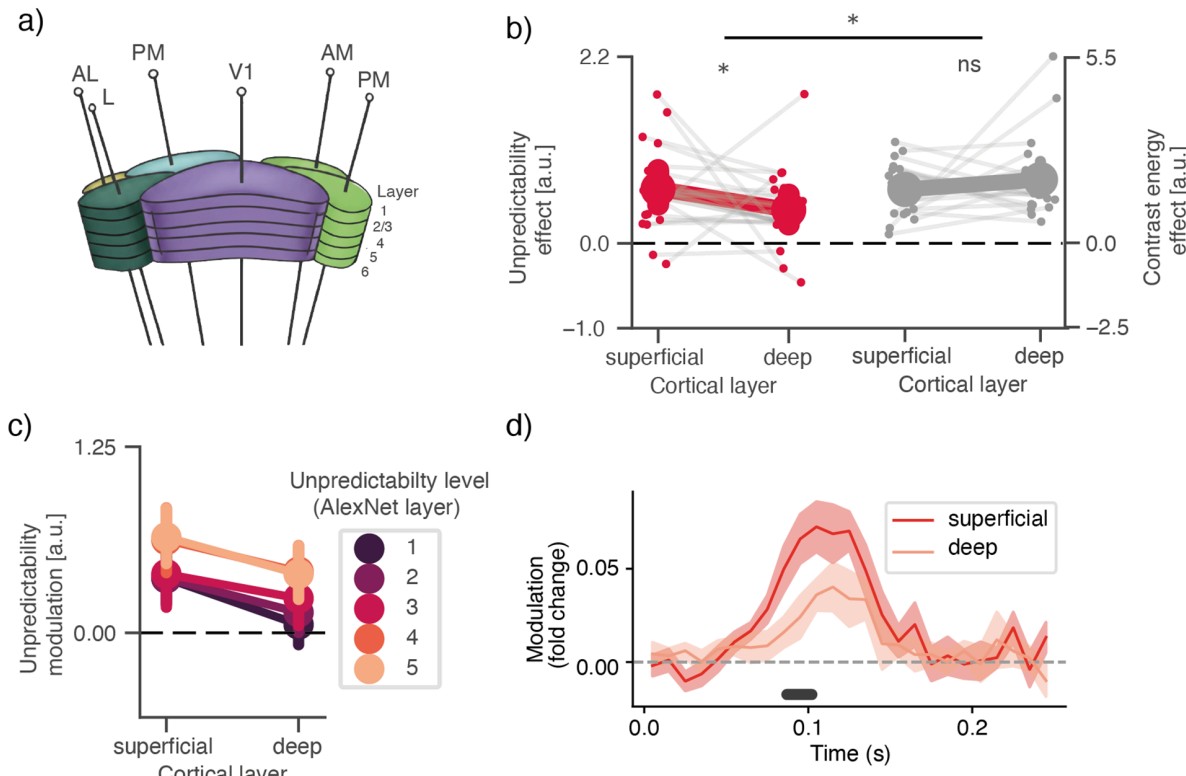

**Fig 4. a)** Neuropixels probe cortical activity across layers of the entire visual system, allowing to dissociate units in superficial (2/3) and deep (5-6) cortical layers. **b)** V1: effect of overall spatial predictability and contrast energy (average coefficient between 75-150ms), in neurons in superficial and deep cortical layers. Small dots with connecting lines indicate individual animals; big dots with error bars indicate mean across animal-averages, and bootstrapped 95% confidence interval. * indicates $p < 0.05$. **c)** Same as in **b**, but for the multi-level predictability estimates; dots indicate mean across animals, error bars indicate 95% confidence interval around the mean. **d)** V1: effect of overall spatial predictability over time (as in Fig 2d), split out for units in superficial versus deep layers. Solid line indicates mean modulation function across animals; shaded area indicates bootstrapped 95% confidence interval around the mean. Black line indicates time points with significant differences (TFCE-corrected $p < 0.05$). For a similar plot for other cortical areas, see S8 Fig.

spatial predictability. Strikingly, in V1, we indeed identified this expected difference: the modulation by spatial predictability was consistently stronger for neurons in superficial layers compared to those in deeper layers (mean summed coefficient difference = 0.336 $FC/\sigma$, 95% CI [0.049,0.605], $p = 0.0173$). Importantly, this laminar difference was specific to predictability and not observed for contrast energy (mean difference = −0.294, 95% CI [−0.949,0.255], $p = 0.2774$; interaction, difference of difference: mean = 0.630, 95% CI [0.054,1.347], $p = 0.011$).

Although the observed laminar difference is subtle, it appears robust across methodological variations, as we found the same patterns not just for overall predictability, but for each of the multi-level predictability estimates (Fig 4c), and across time (Fig 4d), where FWER-corrected comparison reveals significant differences between 85 and 110 ms, showing the effect was not contingent on our specific, pre-defined time-window.

However, because the analysis involved subdividing the already sub-selected units into those classified as 'superficial' and 'deep' (and discarding others), the analysis could only be feasibly conducted in V1; we did not have enough sensitivity in other areas to meaningfully perform the comparison (see S8 Fig).

## 2.4 No effect of short-term experience on predictability modulations

Finally, we asked whether the predictability effects were dependent on short-term experience. One possibility would be that these effects reflect more long-term, 'hard-wired' priors about the structure of the natural visual world, largely independent of recent experience. However, the stimuli were repeated 50 times, so the animals built up extensive experience with the images. Since recent experience and familiarity are known to shape cortical processing in mouse visual cortex [22–24], such extensive repetition might enhance the predictability effect—or may even be required for it to emerge in the first place.

To adjudicate between these possibilities, we estimated the effect separately for the first and second half of the dataset. Unsurprisingly, the average response was lower (Fig 5) in the second half of trials (±25 repetitions), compared to the first (difference between 75-150 ms: mean = 0.076, 95% CI [0.028, 0.119], $p$ = 0.0124), potentially reflecting general adaptation. There was also a small but significant difference for contrast energy (mean = 0.024, 95% CI [0.002, 0.051], $p$ = 0.0182), indicating that contrast energy modulated the neural response less strongly in the second half of trials. Strikingly, however, we observed no difference for the unpredictability effect (mean = 0.004, 95% CI [–0.006, 0.015], $p$ = 0.4006; see Fig 5). Together, this suggests the spatial prediction effects we observed appear independent of recent experience.

## 3 Discussion

We quantified the spatial predictability of local receptive field image patches for thousands of individual neurons using deep generative modelling, and related these predictability estimates to neural responses in a high-quality survey of visual cortex of mice viewing natural scenes. This revealed that neural responses throughout mouse visual cortex are shaped by sensory predictability, even after controlling for a range of low-level image statistics such as local contrast, orientation

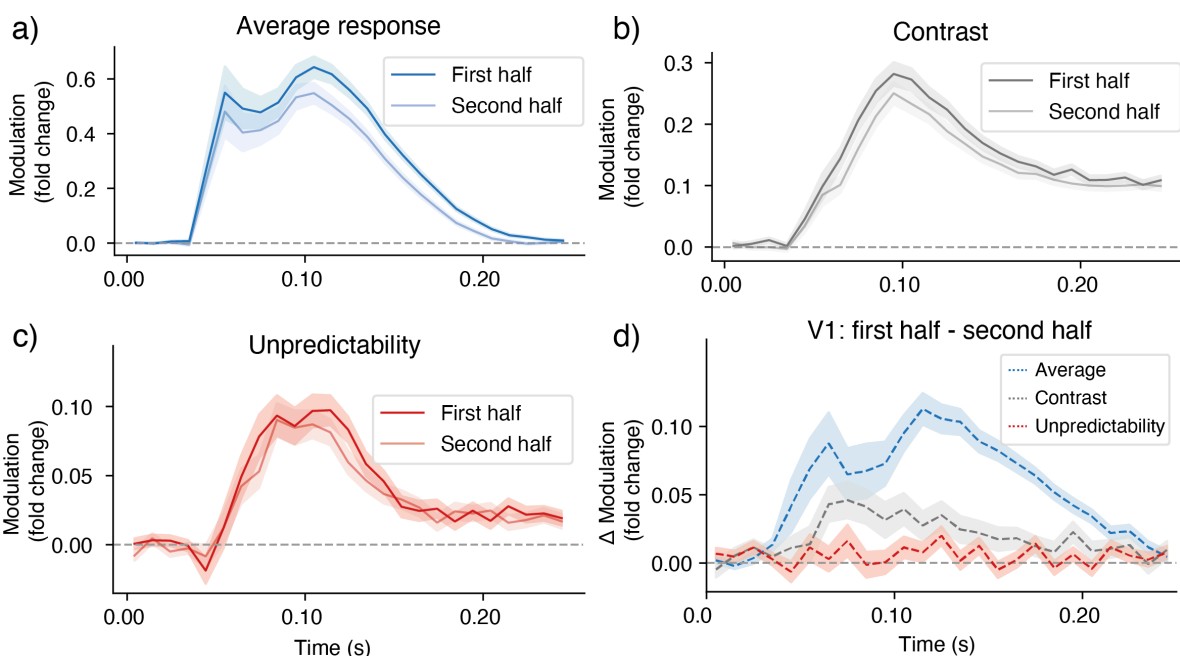

**Fig 5**. **a) Primary visual cortex: average response (coefficient of time-resolved intercept) in first vs second half of the trials. b)** Same but for $\beta$ coefficient of contrast energy in the preferred contrast channel. **c)** Same but for image patch unpredictability. **d)** Difference waves (early-late) for all three effects. More extensive exposure leads to a reduced average response and reduced contrast effect, but no difference for unpredictability. In all panels, solid lines are mean coefficients across the animal-level means, shaded area indicates bootstrapped confidence interval around the across-animal mean.

content, and contour-related properties. In line with predictive coding models, we found that this spatial prediction effect was stronger in superficial layers of primary visual cortex—but contrary to predictive coding models, this prediction error signal is in the wrong frame of reference: neurons are most sensitive to the predictability of higher-level features, even in regions tuned to low-level features. Finally, these effects occurred independent of recent experience with the stimuli, suggesting they reflect longer-term structural priors about the visual world rather than expectations based on the exposure to these specific stimuli. Together, these results provide a new window into how predictive processing operates under natural conditions.

We observe strong modulations of cortical responses by spatial predictability, which suggests that visual cortex is constantly predicting incoming signals. The word "predicting" may seem odd here, as spatial predictions appear quite different from temporal predictions, i.e. predicting an upcoming stimulus, which are typically used to study predictive processing [10,13,16,25]. Note however, that the canonical models that popularised predictive processing [e.g.21,26] deal exclusively with spatial (reconstructive) predictions, and do not include prediction over time. In that way, we study sensory prediction in the traditional predictive processing sense: predicting RF center from RF surround (cf. [27]). What is shared between both types of prediction is the key assumption that the brain compares sensory input to predictions of what signals to expect based on a generative model and the spatial or temporal context. Plus, critically, the neural marker that responses are reduced when incoming signals are more in line with these (top-down or locally recurrent) contextual predictions.

Spatial context has a long history in vision [28–31]. Perhaps the neurophysiologically best characterised spatial context effects come from the center-surround literature. The canonical finding here is *surround suppression*, where responses to stimuli in the RF are reduced by the presence of particular surrounding stimuli [32–34]. This suppression is strongest when the content of the RF is coherent with and therefore predictable by the surround, yielding explanations based on redundancy reduction or predictive coding which cast surround suppression as a kind of expectation suppression, i.e. reduced prediction error [21,35–37]. However, the opposite (surround excitation) is also observed [34,38], and recent modelling-based analyses suggest it may be much more common than previously thought, in particular for natural stimuli [39]. Due to methodological differences, it is difficult to link our results directly to the center-surround effect literature, where the key analysis is based on comparing the content of the RF with the content of the surround, which is not what we do. Conceptually, though, the suppressive effect of predictability (or facilitating effect of unpredictability) we observe is in line with surround suppression, and in particular predictive or efficient coding based explanations thereof. However, they critically depart from typical surround suppression, where modulation often relies on similarly-tuned low-level features. Instead, we find a dissociation between bottom-up feature tuning and predictability tuning, with a dominant modulation by *higher-level feature predictability*, even in areas primarily sensitive to lower-level features. This pattern dovetails with other recent work using deep neural network models and natural stimuli, which has similarly revealed that V1 neurons are tuned to more complex features than traditionally characterized with parametric stimuli [39,40].

Interpreting the magnitude of this effect, however, requires careful consideration. The use of natural scenes is both a strength and a limitation, as it introduces inherent correlations between spatial unpredictability and low-level features like local contrast. This results in a large portion of shared neural variance, meaning that metrics like $\Delta R$, which isolate unique contributions, may systematically underestimate the relative importance of predictability. A clear follow-up would therefore be to use natural images specifically sampled to minimise these correlations (or to maximise the dissociation between different levels of predictability). Furthermore, our approach may underestimate predictability's importance because our metrics are derived from a general-purpose vision model, not one tailored to mouse vision. A rodent-specific model, e.g. a model trained on a 'rodent visual diet', or an experiment using more ecologically valid stimuli might reveal a larger effect, suggesting the estimates reported here may represent a conservative lower bound on the true effect size of spatial predictability.

In our analysis, multiple predictability estimates are computed for each neuron's receptive field. However, this metric does not explicitly account for the specific tuning properties of each neuron. This is a limitation, because a rich body of work using artificial stimuli has established that contextual modulation in the visual cortex can be highly feature-specific,

e.g. being strongest when the surround matches the neuron's preferred orientation [33,41]. Extending this principle to our framework would be a powerful next step. For an orientation-tuned neuron, one could assess not just the overall predictability of an image patch, but specifically the predictability of its preferred orientation content. Such a 'tuning-contingent' metric might reveal stronger and more precise modulations, aligning the analysis more closely with the information each neuron is thought to process. Furthermore, this feature-specific approach would provide a natural framework for modeling the nonlinear dynamics of contextual modulation, for instance, by allowing predictability to directly gate or weigh the divisive influence of surround normalization on a neuron's contrast response.

Predictability modulations were enhanced in superficial layers of cortex (Fig 4). On face value, this may appear an important confirmation of predictive coding theories which critically and uniquely postulate such a laminar dissociation [8,21,42], a distinction for which only recently supporting evidence is starting to accumulate [43–45]. However, we could only reliably observe the effect in V1, and critically, the magnitude of the effect was relatively small. Therefore, if one takes the predictability effect to reflect sensory prediction errors, this could even appear to speak against a strong laminar segregation, where prediction errors are only encoded in superficial layers. However, we caution against drawing strong conclusions in either directions, because the neural markers of prediction errors and the downstream effects of prediction errors are difficult to disentangle. Even in a classical predictive coding circuit—with error units in superficial and prediction units in deep layers—stronger prediction errors would result in stronger updates in prediction units, and therefore, in the aggregate, more spiking activity for more unpredictable stimuli in deep layers too. Therefore, while the laminar effect we observe is striking, strong conclusions about the existence of error/prediction units would require more granular analyses at both theoretical and physiological levels. Theoretically, simulations of predictive coding models using matched stimuli and receptive field configurations would provide more precise laminar predictions—though implementing such models for natural scenes remains an open challenge. Physiologically, larger samples of simultaneously recorded neurons per layer and histological validation of laminar boundaries would reduce the uncertainty inherent in the current, coarser, classification method.

The predictability effects we observe appeared surprisingly independent of recent experience. This is notable because cortical processing in mouse visual cortex is typically highly malleable, with neural responses strongly shaped by recent experience and familiarity [22–24]. This stability particularly contrasts with the findings of [46], who, studying similar spatial context effects, found that responses to occluded scenes strengthen systematically as mice learn the complete versions of these images. These divergent findings reveal that even for spatial prediction, mouse visual cortex implements multiple prediction mechanisms in parallel: while some contextual effects rely on rapid learning of specific patterns, others appear to reflect more stable structural priors about the visual world.

Perhaps most strikingly, we observed a divergence between feature tuning and predictability tuning: Neurons are most sensitive to the predictability of higher-level visual features, even in areas most sensitive to low-level features, like V1 (Fig 3). This is a departure from classical predictive processing models, where the content of predictions aligns with the tuning for visual features, via hierarchical inference. For instance, early stages would reflect expected low-level features based on higher-order interpretations, such as when a faint edge is highly expected as a part of a key object boundary [21,26]. However, despite departing from theoretical models, our results align with emerging empirical evidence. The closest parallel is from [3] who performed a similar analysis in macaque V1 and also found greatest sensitivity to higher-level feature predictability. This was further corroborated by [20] who used representational similarity analysis (RSA) to examine the content of temporal predictions in a more classical, controlled experiment of visual stimuli predictably following other stimuli. Despite their distinct methodological approach, they also observed preferential encoding of higher-level predictability throughout the visual hierarchy. The convergence of these findings across multiple species (mouse, macaque, human) and experimental approaches (electrophysiology, fMRI) appears to point to a fundamental functional principle: visual cortex predicts sensory information primarily at higher levels of abstraction, even in early visual areas.

Why would visual cortex predict sensory input at higher levels of abstraction? A tantalising hint comes from recent advances in machine learning. There, researchers tried to develop general vision models by training networks to predict pixels in context [47,48]—much in the same way as LLMs can learn general models of language by predicting words in context [49,50]. However, initial attempts in vision were not nearly as successful as in language: While these models become effective at predicting pixels, they were not so effective for learning general representations that can be applied to other tasks (e.g. object recognition). Recently however, new techniques are starting to show success [51–56]. What these methods have in common is that they encourage predicting higher-level visual features instead of lower-level visual features. The intuition here is that—contrary to text—images contain very high levels of redundancy of low-level information, making low-level prediction tasks too simple to drive meaningful learning. By focusing on higher-level features instead, models are forced to develop more abstract and useful visual representations [51,55]. Our results suggest the brain may have found a similar strategy, predicting specifically at a higher level of abstraction, perhaps as a way for effective self-supervised learning of visual representations.

Altogether, our results demonstrate that predictive processing is pervasive throughout visual cortex, operating predominantly at higher levels of abstraction - even in early sensory areas. This organization may reflect a fundamental neural strategy for learning robust visual representations from the statistical structure of the natural world.

## 4 Methods

### 4.1 Stimuli and data

We analysed the 'Brain Observatory 1.1' subset of the Allen Institute *Visual Coding Neuropixels* dataset [1]. This is an openly available dataset that surveys spiking activity across the mouse visual system, recorded using 6 simultaneously inserted Neuropixels probes. Mice viewed a range of full-field stimuli, presented monocularly to the right eye. We focus on two stimulus sets: Gabors—for Receptive Field (RF) mapping—and Natural Scenes. The latter comprises 118 scene images, presented for 250 ms, for 50 repetitions, resulting in 5900 trials (Fig 1A).

For analysis, raw spike times were binned into 10 ms intervals to compute time-resolved firing rates. To enhance signal-to-noise ratio, firing rates were smoothed across time using a uniform moving average filter with a 30 ms window. Because different neurons exhibited widely varying firing rate ranges, we normalised the firing rates to decrease the variance between neurons and avoid that the population or animal-level averages would be dominated by high-firing-rate neurons, We normalised firing rates as fold change relative to baseline, defined as the mean activity during the initial 20 ms of stimulus presentation. This normalisation was computed as (FR-baseline)/baseline, with a small pseudo-baseline value (of 0.5 Hz) added to avoid division by extremely small numbers.

We further enhanced the signal-to-noise ratio by averaging responses across multiple presentations of the same stimulus. Specifically, the 50 repetitions of each natural scene were divided into 5 chunks, with each chunk representing the average response to approximately 10 presentations. This chunking approach preserved stimulus-specific response patterns while reducing noise, providing more reliable inputs for subsequent regression analyses. Finally, to minimise the influence of eye movements and blinks, we implemented quality control on each trial. Trials were excluded (i.e. removed before the averaging into chunks) if they contained blinks, or outlying horizontal or vertical eye positions, defined as those exceeding 2 standard deviations from the mean.

### 4.2 Receptive field mapping and unit selection

RFs were modelled by fitting a 2d-Gaussian on the smoothed 2d spike histograms of the positions where gabors were presented. This resulted in RF for every identified unit. Subsequently, we selected all units for which this ideal RF that was on-screen and had a size between 150 and 750$°^2$; we did this because for much larger RFs the spatial predictability analysis did not work as the RFs became too large to inpaint. Moreover, we required that the "ideal" RF was a good fit, i.e. that the raw (smoothed) 2d histograms could be well approximated by the gaussian ($\rho > 0.7$). After applying these RF

criteria and combining with the quality assurance criteria reported in [1], this resulted in 3176 analysed cortical neurons in total that were located at the 6 pre-defined cortical areas of interest: 1173 neurons for V1, 354 neurons for VISl, 364 neurons for VISrl, 525 neurons for VISal, 253 neurons for VISpm, and 507 neurons for VISam. For each unit, we used its recorded Common Coordinate Framework (CCF) coordinates to putatively identify its corresponding cortical layer in the Allen Mouse Brain Atlas. Units were classified as "superficial" if they resided in cortical putative layers 1-3 and "deep" if they resided in putative layers 5-6.

## 4.3 Spatial predictability inpainting model

Our inpainting model employed a PConvUNet architecture with partial convolutions, following Liu et al. [2]. The encoder consisted of up to eight blocks, with the first using a 7×7 kernel followed by 5×5 kernels in the second and third blocks, and 3×3 kernels in subsequent blocks. Each encoder block combined a partial convolution with batch normalization (except the first layer) and ReLU activation. The partial convolution technique progressively filled missing regions by updating a binary mask after each operation, normalising feature values based on the ratio of valid pixels.

The decoder mirrored the encoder with eight blocks, each containing nearest-neighbor upsampling followed by concatenation with corresponding encoder features, partial convolution, batch normalization, and LeakyReLU activation. This skip-connection structure preserved spatial information lost during downsampling. During fine-tuning, encoder batch normalization layers were frozen to maintain pretrained feature extraction capabilities while allowing decoder parameters to adapt to the new dataset characteristics.

The loss function combined reconstruction, content, and style components with weighted hyperparameters. Reconstruction loss incorporated Fourier domain metrics, while content and style losses compared VGG-16 activations and their Gramian matrices, respectively, minimising perceptual artifacts common in generative models.

## 4.4 Model training and fine-tuning

We used a pretrained model initialized with VGG-16 weights and first trained on the Places2 dataset [57] for 60k iterations with a batch size of 64, focusing on large circular masks to improve performance on receptive field-sized regions. We employed the Adam optimizer with a learning rate of 0.0005 and default momentum parameters ($\beta_1 = 0.9$, $\beta_2 = 0.999$).

For the final adaptation stage, we fine-tuned the model for 1500 steps on the Van Hateren Natural Images dataset [6], using a learning rate of $9.5e{-}5$, excluding images used in the Allen Institute Dataset. This step specifically optimized the model for grayscale natural image statistics matching our experimental stimuli.

We adapted the loss function from the implementation of Liu et al. [2], which combined multiple weighted components:

$$L_{total} = L_{valid} + 12 \cdot L_{hole} + 0.1 \cdot L_{TV} + 0.05 \cdot L_{perceptual} + 120.0 \cdot L_{style}$$

where $L_{valid}$ and $L_{hole}$ are per-pixel reconstruction losses on non-masked and masked regions respectively, $L_{TV}$ is the total variation loss, $L_{perceptual} = \sum_{\ell} \|\phi_\ell(y) - \phi_\ell(\hat{y})\|_2^2$ measures content across VGG layers, and $L_{style} = \sum_{\ell} \|G_\ell(y) - G_\ell(\hat{y})\|_1$ compares Gram matrices $G_\ell = \phi_\ell^T \phi_\ell / C_\ell$ capturing texture statistics. Following Liu et al.'s implementation, we used these loss coefficients during fine-tuning to balance structural accuracy and perceptual quality. Complete mathematical derivations of individual loss components are provided in Liu et al. [2].

## 4.5 Spatial predictability analysis

For each neuron, we defined an RF mask as the area within the full-width at half-maximum of the gaussian receptive field. This mask was removed from the image and then in-painted by the model. For each image, the filled-in (predicted) RF patch was compared with the actual patch to compute predictability (Fig 1A), for this we took a rectangular mask spanning the RF circle with a dimension of $\sqrt{1.5}$ times the diameter of the RF. Unpredictability was defined as the $\ell_2$ distance

between the observed and predicted patches at each convolutional layer of AlexNet, a shallow CNN with a good fit to mouse visual cortex [19]. We compute the discrepancy between predicted and actual patch in the feature space of a deep network because it captures perceptually relevant differences while remaining insensitive to pixel-level artifacts or distortions such as noise and intensity variations that don't affect perceptual content, thus providing a more robust measure of perceptual distance and hence predictability [58].

In our analyses, we use two types of spatial predictability estimates: the overall spatial predictability (e.g. in Fig 2 and elsewehere where indicated), and multi-level spatial predictability. The overall spatial predictability is meant to serve as an omnibus metric, for the research questions where we were not interested in feature-specific predictability. To reduce researcher degrees of freedom, we simply defined it as the average of the $\ell^2$ distance for the individual layers. The multi-level predictability estimates, by contrast, where simply the layer-specific $\ell 2$ distances. Because early CNN layers extract simple features (e.g. edges) and higher levels more complex ones (e.g. textures), performing the comparison at each layer allows to quantify predictability at multiple levels of abstraction (Fig 3a).

While the predictability estimates at different levels are correlated, they also partially and systematically dissociate. Post-hoc inspection confirms they dissociate in predictable and interpretable ways. Some key illustrative examples are included in Fig 3. Perhaps the most common stimulus patch that dissociates high and low-level predictability are textures, where the exact spatial configuration of low-level features (edges, orientations, and spatial frequencies) is highly unpredictable, but the higher-level statistical structure is perfectly predictable. This phenomenon aligns with the seminal work of Portilla and Simoncelli [59], who demonstrated that textures can be mathematically defined and synthesised by matching higher-order correlations among low-level features (while having the low-level features themselves random and unpredictable). This principle explains why texture regions in our dataset occupy the upper-right quadrant in Fig 3B, exhibiting high low-level unpredictability but low high-level unpredictability. At other patches, both the high and low-level features are highly unpredictable, typically occurring at boundaries between distinct objects or scenes where contextual information from the surroundings provides insufficient constraints to predict either low-level features or higher-order structure within the receptive field.

### 4.6 Control variables

To control for low-level statistics extracted in the feedforward sweep, we implemented a biologically-inspired contrast model that captures how early visual neurons respond to images, by computing both first-order and second-order contrast [4,5,60]. The model computes local contrast using quadrature-phase Gabor filters [61], which mimic the orientation and spatial frequency selectivity of V1 simple cells. Specifically, we constructed two filter banks: a first-order or contrast energy (CE) bank with spatial frequencies spanning 5 octaves (0.02-0.32 cycles per degree) and a second-order contrast or spatial coherence (SC) bank with slightly lower frequencies (0.015-0.24 cycles per degree). For each filter bank and spatial frequency, we computed the energy response $E(x,y)$ by combining quadrature-phase filter responses $F_0$ and $F_{\pi/2}$ according to $E(x,y) = \sqrt{F_0(x,y)^2 + F_{\pi/2}(x,y)^2}$. This energy-based formulation quantifies contrast along specific orientations (8 orientations, spanning $0$-$\pi$ radians). We then applied divisive normalization to these energy maps using local statistics: $E_{norm}(x,y) = \frac{E(x,y) \cdot \max(E)}{E(x,y) + \max(E) \cdot S(x,y)}$, where $S(x,y)$ represents the local coefficient of variation (standard deviation divided by mean) computed over a Gaussian window. This step models the suppressive surround observed the visual system and enhances the representation of contrast boundaries.

From these normalised contrast maps, we extracted several summary statistics for each neuron's receptive field across all 118 natural scenes. Due to the limited number of unique images, we could not fit the contrast energy of an entire gabor pyramid to the neural response, and instead had to capture relevant contrast information with a minimal set of regressors. For each neuron, we first identified its preferred spatial frequency and orientation from the Gabor stimulus responses. We

then computed the following sets of metrics within each receptive field: (1) five first-order contrast energies at five spatial frequency scales (0.02-0.32 cpd), representing the average normalised energy within the receptive field; (2) five second order contrast statistics (SC) at corresponding scales (0.015-0.24 cpd), computed as the inverse coefficient of variation ($\mu/\sigma$) of the receptive field of the SC image; (3) preferred contrast energy, capturing energy specifically at each neuron's preferred spatial frequency and orientation; (4) circular variance, which quantifies orientation homogeneity within the receptive field, allowing us to capture neural responses to extended contours [62]; (5) relative orientation energies, indexing contrast at orientations $\pm30°$ from the preferred orientation of the specific neuron, averaging across octaves; and (6) root-mean-square (RMS) contrast within the receptive field. Additionally, we incorporated running speed as a non-visual control variable, averaged across trial chunks, to account for locomotion-dependent modulation of visual responses. Altogether, this comprehensive set of regressors enabled us to control for low-level feedforward processing while maintaining a statistically tractable model.

## 4.7 Regression analysis

To assess the effect of spatial predictability on neural responses, we performed time-resolved regression analyses on the normalised firing rates. For each neuron $i$ and time point $t$, we fitted separate linear regression models to the normalised firing rate across all observations:

**Baseline model:**

$$\mathbf{y}_{i,t} = \mathbf{X}_{i,\text{baseline}}\beta_{\text{baseline}} + \epsilon \tag{1}$$

**Extended model:**

$$\mathbf{y}_{i,t} = \mathbf{X}_{i,\text{extended}}\beta_{\text{extended}} + \epsilon \tag{2}$$

where $\mathbf{y}_{i,t}$ is the vector of neural responses across observations, and the design matrix $\mathbf{X}_{i,\text{baseline}}$ contains the intercept plus baseline variables

$$\mathbf{X}_{i,\text{baseline}} = \Big[\mathbf{1} \mid \text{CE}_1 \mid \text{CE}_2 \mid \text{CE}_3 \mid \text{CE}_4 \mid \text{CE}_5 \mid \text{SC}_1 \mid \text{SC}_2 \mid \text{SC}_3 \mid \text{SC}_4 \mid \text{SC}_5 \mid$$
$$\text{CE}_{\text{pref}} \mid \text{CV} \mid \text{CE}_{\text{pref}-30} \mid \text{CE}_{\text{pref}+30} \mid \text{RMS} \mid v\Big] \tag{3}$$

The extended model augments this with spatial unpredictability: $\mathbf{X}_{i,\text{extended}} = [\mathbf{X}_{i,\text{baseline}} \mid P]$. Here, $\text{CE}_k$ denotes first-order contrast energy at spatial frequency scale $k$ ($k = 1, \ldots, 5$ corresponding to 0.02–0.32 cycles per degree), $\text{SC}_k$ denotes second-order contrast (inverse coefficient of variation, $\mu/\sigma$) at scale $k$ (0.015–0.24 cpd), $\text{CE}_{\text{pref}}$ is the contrast energy at neuron $i$'s preferred spatial frequency and orientation, CV is circular variance, $\text{CE}_{\text{pref}-30}$ and $\text{CE}_{\text{pref}+30}$ represent contrast energies at orientations $-30°$ and $+30°$ relative to the preferred orientation (averaged across spatial frequency octaves), RMS is root-mean-square contrast, $v$ is running speed, and $P$ is the spatial unpredictability metric. All spatial features were computed within neuron $i$'s receptive field (and neuron-specific tuning properties) resulting in a unique design matrix $\mathbf{X}_i$, and separate model estimation, for each neuron. Individual regression models were estimated for each neuron at each timepoint (10 ms bins), allowing us to track the temporal evolution of predictability effects throughout the visual response. We constructed separate models for each predictability metric rather than including multiple predictability estimates simultaneously, since the correlations between different predictability metrics were often high, complicating the interpretation of the coefficients and $\Delta R$.

We employed two complementary regression approaches. First, we estimated the full model coefficients ($\beta$) to perform inference on the relationship between predictability and neural responses. This allowed us to quantify the magnitude and timing of predictability effects across different visual areas and cortical layers. Because in every regression we fit, all predictor variables were normalised (z-scored), the resulting standardised $\beta$ coefficients from these models revealed how

strongly neural activity was modulated by predictability compared to low-level features. Second, we used cross-validation to evaluate model performance. For each neuron, we implemented a shuffle-split cross-validation procedure (10 splits with 75% training, 25% testing) to compute correlation coefficients between predicted and actual neural responses. By comparing the performance of models with and without the predictability term, we calculated a difference in correlation coefficients ($\Delta R$), which quantified the unique contribution of predictability to explaining neural responses beyond what could be accounted for by low-level image statistics and behavioural variables alone.

### 4.8 Encoding-based feature sensitivity analysis

To systematically quantify feature sensitivity across visual cortical areas, we implemented an encoding approach that assessed the predictive power of different feature levels on neural responses. Following [19], we employed a pretrained AlexNet CNN to extract hierarchical visual features from the stimulus set. For each unit, we constructed encoding models using feature maps from each AlexNet layer, trained to predict trial-averaged neural responses to natural scenes, with data partitioned into chunks to enhance signal-to-noise ratio while preserving stimulus-specific patterns. To accomodate the high dimensionality of CNN features relative to our stimuli (118 scenes), we employed Partial Least Squares regression with 25 components, capturing predictor space variance while avoiding overfitting. Model performance was evaluated using shuffle-split cross-validation (10 splits, 75% training, 25% testing) to compute correlation coefficients between predicted and actual neural responses, generating layer-specific encoding accuracies for each unit.

Because our primary interest was in comparing relative preferences for stimulus features, we normalizeised cross-validated correlation scores by dividing by the maximum correlation achieved across layers for each neuron. This prevented high-SNR units from dominating population-level analyses while preserving relative layer preference patterns. The resulting normalised profiles enabled direct comparison of feature selectivity across different visual areas and cortical layers, providing complementary evidence to our spatial predictability analysis.

### 4.9 Statistical testing

Statistical testing was performed using multi-level inference, first computing averages across units in one animal, and then performing statistics across animals. For simplicity, we ignored the fact that different animals had different amounts of (selected) neurons per region, and weighted the contribution of each animal equally.

To compute p-values we used data-driven, non-analytical bootstrap t-tests, that involve resampling a null-distribution with zero mean (by removing the mean), counting across bootstraps how likely a t-value at least as extreme as the true t-value was to occur. Each test used at least $10^4$ bootstraps; p-values were computed without assuming symmetry (equal-tail bootstrap; [63]). Confidence intervals (in the figures and text) are also based on bootstrapping, with $10^4$ resamples.

For time-resolved paired comparisons, we corrected for multiple comparisons using a temporal clustering permutation test. This was implemented using the Threshold-Free Cluster Enhancement (TFCE) algorithm, which integrates cluster size and height information across a range of thresholds [64]. The analysis was performed using the implementation and all default parameters in MNE-Python [65].

### 4.10 Synthetic stimuli validation

To validate the predictability metric, we generated two sets of synthetic parametric stimuli (384x384 pixels). The first set, simulating classical end-stopping (S1 Fig), consisted of a horizontal bar composed of fine-grained alternating stripes. The bar was presented in three configurations relative to a central circular mask: confined within the mask ('No Context'), extending 20% beyond it ('Some Context'), or spanning the full image width ('Full Context'). The second set, designed to dissociate feature-level predictability (S2 Fig), used a constant vertical grating as the surround. The central masked region was filled with one of three ground-truth patterns: a continuous vertical grating (no mismatch), an orthogonal grating ('Shifted'), or a grating with added phase noise ('Noisy'). Since the model's input (the masked image) was identical

for these three conditions, it produced a single reconstruction, allowing us to measure how unpredictability changed as a function of the specific mismatch between the reconstruction and each ground truth. In all cases, the computed loss was normalised (by dividing across the max loss of each layer) to ensure the content losses are at the quantitative same scale for visualisation.

## Supporting information

**S1 Fig. Inpainting-based unpredictability qualitatively recapitulates classical end-stopping.** To confirm that our pre-dictability metric is sensitive to canonical forms of spatial context, we simulated a classical end-stopping experiment. A) Three synthetic stimuli were generated where a central receptive field (RF) patch (red dashed circle, corresponding to the in-painted area) was presented with a horizontal line segment. The line was either confined to the RF patch ('No Context'), extended slightly into the surround ('Some Context'), or extended far into the surround ('Full Context'). B) We computed multi-level unpredictability for the RF patch, quantified as the normalized content loss between the actual and predicted patch features from the first five convolutional layers of AlexNet. Unpredictability systematically decreased as more context was provided, consistent with the suppressive effect observed in end-stopped neurons. C) The magnitude of this contextual modulation, computed as the fold-change reduction in unpredictability from the 'No Context' to the 'Full Context' condition. The effect was feature-specific and strongest for low-level features (Conv2), providing a proof-of-concept that the inpainting-based modelling analysis, designed for natural stimuli, captures fundamental principles of contextual processing with simple stimuli too.
(TIFF)

**S2 Fig. Dissociating feature-level predictability with targeted synthetic mismatches.** This analysis demonstrates how the multi-level unpredictability metric is sensitive to different feature violations (see Methods). A) We created three ground-truth stimuli where the surround was a continuous vertical grating. The central masked patch contained either a continuous grating (no mismatch), an orthogonal grating ('Shifted'), or a grating with phase noise ('Noisy'). Because the masked input to the inpainting model was identical in all cases, the model produced the same reconstruction: a continuous vertical grating. B) Unpredictability was quantified as the normalised content loss between each ground truth and the single reconstruction. This revealed a clear dissociation: low-level unpredictability (Conv1) was maximally driven by the orientation mismatch in the 'Shifted' condition. In contrast, high-level unpredictability (Conv5) was much less sensitive to the orientation shift but strongly driven by the textural disruption in the 'Noisy' condition. Between Conv1 and Conv5 we observe a gradual shift from being most sensitive to orientation disruption (shifted grating) towards most sensitive to textural disruption (phase noise). This confirms that the metric can distinguish between violations of low-level and higher-level image properties.
(TIFF)

**S3 Fig.** Example patches for each of the four quadrants of feature-level predictability estimates. For each of the three examples per quadrant, the figure shows the input patch, the masked version, and the model's prediction. The quadrants illustrate The quadrants illustrate intuitive dissociations between low- and high-level unpredictability. For instance, patches with low unpredictability on both axes often contain simple, predictable structures like edges, while patches with high low-level but low high-level unpredictability are typically textures. Conversely, high unpredictability at both levels is often observed for complex objects fully spanned by the receptive field. The rarer case of low low-level but high high-level unpredictability can occur for complex objects that are primarily composed of sharp, predictable edges, or other complex objects that contain simple edges. See Methods for more information. All natural images from [1,6].
(TIFF)

**S4 Fig. Multi-level predictability analysis for all visual cortical areas.** Same as red line in Fig 3 but for all cortical areas. Interestingly, all visual areas show a negative relationship, where sensitivity is highest to the lower-level features. However, in hierarchically higher cortical areas the relationship is more shallow, with less of a preference for lower level features, and lower encoding performance and more variance across units and animals. Dots show mean across animal-level averages; error bars show bootstrapped 95% confidence intervals around the mean. (TIFF)

**S5 Fig. Multi-level feature sensitivity analysis for all visual cortical areas.** Same as blue line in Fig 3 but for all cortical areas. Interestingly, all visual areas show a similar positive relationship, where sensitivity is highest to the predictability of higher-level features, and lowest for to the predictability of lower-level features. However, in hierarchically higher cortical areas (such as VISam), the relationship appears steeper, so there is a stronger preference for higher-over-lower level features. Dots show mean across animal-level averages; error bars show bootstrapped 95% confidence intervals around the mean. (TIFF)

**S6 Fig. Multi-level feature sensitivity analysis based on $\Delta R$. a)** V1. Same as red line in 3c, but using the cross-validated $\Delta R$ instead of the time-averaged coefficient as a metric of interest. **b)** $\Delta R$-based unpredictability tuning analysis for all visual areas of interest. X-ticks indicate cortical area, colour indicates feature space or unpredictability level. Just like in the coefficient-based analysis (Figs 3 and S5) we see the same positive relationship in all areas, where the unpredictability sensitivity highest for high-level unpredictability, and lowest for low-level unpredictability. Dots show mean across animal-level averages; error bars show bootstrapped 95% confidence intervals around the mean. (TIFF)

**S7 Fig. Control analysis with synthetic noise images shows no intrinsic trend for unpredictability.** To confirm that the preference for high-level unpredictability (Fig 3c) reflected the neural response to images rather than an inherent property of our method, we repeated the analysis using synthetic noise images. Unpredictability sensitivity is computed as the time-averaged coefficient. Dots show the mean across mice with 95% confidence intervals. No positive trend or any significant effect of unpredictability was observed. (TIFF)

**S8 Fig. Laminar comparison of overall spatial predictability across cortical areas.** Same as Fig 4d, but for all cortical areas: lines with shaded error bars show the coefficient of overall spatial predictability over time, split out for units classified as 'superficial' or 'deep'. Note that this sub-splitting of already sub-selected units reduces the number of neurons, and the number of animals with enough units in both categories. (TIFF)

**S9 Fig. Latencies across ROIs and predictability metrics.** Big dots with error bars indicate mean latency (time to 50% of max), plus bootstrapped 95% confidence intervals across mice. (TIFF)

## Author contributions

**Conceptualization:** Micha Heilbron, Floris P. de Lange.

**Formal analysis:** Micha Heilbron.

**Funding acquisition:** Floris P. de Lange.

**Investigation:** Micha Heilbron.

**Methodology:** Micha Heilbron.

**Project administration:** Micha Heilbron.

**Resources:** Micha Heilbron.

**Software:** Micha Heilbron.

**Supervision:** Floris P. de Lange.

**Validation:** Micha Heilbron.

**Visualization:** Micha Heilbron.

**Writing – original draft:** Micha Heilbron.

**Writing – review & editing:** Micha Heilbron, Floris P. de Lange.

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
