## [Decision Letter · Decision Letter 0]

28 Aug 2025

PCOMPBIOL-D-25-00957

Higher-level spatial prediction in natural vision across mouse visual cortex

PLOS Computational Biology

Dear Dr. Heilbron,

Thank you for submitting your manuscript to PLOS Computational Biology. After careful consideration, we feel that it has merit but does not fully meet PLOS Computational Biology's publication criteria as it currently stands. Therefore, we invite you to submit a revised version of the manuscript that addresses the points raised during the review process.

Please submit your revised manuscript within 60 days Oct 28 2025 11:59PM. If you will need more time than this to complete your revisions, please reply to this message or contact the journal office at ploscompbiol@plos.org. Please include the following items when submitting your revised manuscript:

We look forward to receiving your revised manuscript.

Kind regards,

Xue-Xin Wei

Academic Editor

PLOS Computational Biology

Marieke van Vugt

Section Editor

PLOS Computational Biology

**Journal Requirements:**

Potential Copyright Issues:

i) Figure 1. Thank you for indicating that “natural images from the Allen Institute Visual Coding Neuropixels dataset.” Please include in the figure legend a direct link to the source of the images and a provide a link to the terms of use / license information.

ii) Please confirm (a) that you are the photographer of 3, or (b) provide written permission from the photographer to publish the photo(s) under our CC BY 4.0 license.

ii) Figure 4 . Thank you for stating that it is "adapted from Siegle et al ". Please provide a link to the terms of use / license information.

Note : If the figure is adapted from a copyrighted source, please provide written permission from the copyright holder to publish this under our CC-BY 4.0 license, or remove the figure / replace the image. Please note we do not recommend using standard request forms available on Publishers' websites, as they grant single use rather than republication under an open access license.

5) In the online submission form, you indicated that your data will be submitted to a repository upon acceptance. We strongly recommend all authors deposit their data before acceptance, as the process can be lengthy and hold up publication timelines. Please note that, though access restrictions are acceptable now, your entire minimal dataset will need to be made freely accessible if your manuscript is accepted for publication. This policy applies to all data except where public deposition would breach compliance with the protocol approved by your research ethics board. If you are unable to adhere to our open data policy, please kindly revise your statement to explain your reasoning and we will seek the editor's input on an exemption.

2) If any authors received a salary from any of your funders, please state which authors and which funders.

7) Please ensure that grant recipient match between the Financial Disclosure field and the Funding Information tab in the online submission form.

8) Please provide a completed 'Competing Interests' statement, including any COIs declared by your co-authors. If you have no competing interests to declare, please state "The authors have declared that no competing interests exist". Otherwise please declare all competing interests beginning with the statement "I have read the journal's policy and the authors of this manuscript have the following competing interests:"

**Reviewers' comments:**

Reviewer's Responses to Questions

Reviewer #1: ### **Summary**

This manuscript presents a compelling test of whether spatial predictability of image patches modulates firing rates in mouse visual cortex. By training inpainting U-Nets to reconstruct missing image regions, the authors derive a measure of “unpredictability” within each neuron’s receptive field and include it as an additional regressor in predicting neural responses.

Using this novel method, the authors answered a few very interesting scientific question. Key results are 1) **Unpredictability explains neural variability** beyond classical visual features. 2) **V1 neurons are more sensitive to high-level unpredictability** than to low-level unpredictability. 3) **This effect is slightly stronger in superficial layers** of V1. 4) **Contrast adaptation vs. unpredictability sensitivity**: Stimulus repetitions alter contrast modulation strength (adaptation) but leave unpredictability sensitivity unchanged, suggesting a more stable, long-term role for unpredictability.

Overall, the regression analyses are rigorous and well controlled, making the results—and even the null findings— convincing. The discussion linking higher-level feature prediction to recent advances in self-supervised learning in latent space is insightful. I recommend acceptance pending minor revisions.

### **Major Issues**

- **Opposing trends in feature predictivity vs. unpredictability modulation**

Figure 3C highlights intriguing, opposite trends between feature predictivity and unpredictability modulation. To ensure these trends are not inherently anticorrelated (e.g., due to differences in scale or variance across feature layers), the authors should consider including a synthetic control. For example, apply the same regression pipeline to random pixel functions or to activations of random neural-network units—this would demonstrate that the observed anticorrelation is not a mere result of the features, but something to do with neuronal tuning..

- One concrete question is: Are regression coefficients directly comparable across different AlexNet layers, or might layer-specific variances drive these differences?

- **Statistical robustness of the depth–predictivity relationship (Figure 4A)**

The depth–predictivity relationship appears noisy, with several flat or reversed trends. The authors should apply alternative statistical tests to confirm its robustness.

- In terms of plotting style, in Figures 4B–C, large markers occlude underlying data. Consider reducing marker size or adding transparency to reveal individual points and trend lines.

- If the result does not achieve statistical significance, the authors should temper the language in Section 2.3’s title.

- **Use of** $\Delta R^2$ **to quantify unique contributions**

The manuscript reports changes in correlation ($\Delta R$), but correlation coefficients are not really additive. A more rigorous metric is the change in explained variance ($\Delta R^2$), which directly reflects each variable’s unique contribution. Please compute and report $\Delta R^2$ to verify consistency with the $\Delta R$ findings. I guess most of them should be the same.

- **Interpretation of the Split-Half Analysis Results**

We suggest refining the title and framing of Section 2.4 to emphasize that repetition-driven adaptation is more parsimoniously explained by modulations in contrast sensitivity than by changes in unpredictivity.

Unpredictivity, as computed here, depends solely on the stimulus image and the neuronal receptive field. Just like classical image features (e.g., contrast, spatial frequency), unpredictivity remains invariant across repeated presentations of the *same* image. Consequently, any repetition-related change in the regression weights must reflect an adjustment in the *relative* strength of these features. So the analysis of using the same unpredictivity measure to regress first and 2nd split of data seems to be misaligned with the goal of examining short term effect on predictivity modulation. Could it be that the predictivity metric itself changed in the first and 2nd half due to exposure?

Therefore, we recommend tempering the language in the current Section 2.4 title (and corresponding conclusions). The authors could clarify this in their response, it’s possibly my confusion.

### **Minor Concerns**

- Captions for Figures S1 and S2 appear mismatched or contain typos; please correct them. As a side note, I also found the trend across visual hierarchy quite interesting, authors could consider move some of them to Figure 3.

- In the Methods, authors can include explicit regression equations to improve clarity.

- For Figure 2, clarify whether inputs (contrast energy, unpredictability) were normalized (e.g., z-scored) before regression; without normalization, beta coefficients cannot be interpreted as the modulation strength as the authors did.

Reviewer #2: This work develops a computational method that investigate the neural mechanisms of predictive vision in mice, following up on earlier primate work. Using the Allen Visual Coding Dataset, the authors demonstrate that predictive processing principles seen in macaque V1 also hold in mouse VISp, including higher-order visual areas, in laminar resolution. The methods are clearly explained and the analyses and controls are in place. My suggestions are minor—mainly simplifying some wording to improve readability and, if possible, visualizing a few extra insights already covered in the results / discussion sections. Overall, I highly recommend the manuscript for publication.

I have one major suggestion, which is to simplify the working around “predictability”. Predictive coding terminology is already quite loaded and “predictability” tends to mean different things to different people, like spatial or temporal predictability. It would be very useful if the definition is clarified at the beginning of the results section (or introduction), so the reader won’t rely on priors (and the rest of the text follows that definition of predictability). I find that “predictability, unpredictability, multi-level predictability, low/high-level predictability” are used in place and clear. However, there are multiple phrases that refer to the same definition of “predictability”:

Predictability, spatial predictability, visual spatial predictability, stimulus predictability, image predictability, image patch predictability, image feature predictability - and they are sometimes used one after the other, which likely confuses the reader. Some instances are:

Section 1 introduces spatial/stimulus predictability,

Figure 1 caption use both spatial predictability, image patch predictability,

Results Section 2.1 highlight image patch predictability, used in section 2 text, Figure 4, Supp. Figure captions

Section 2.2 title: Image predictability

Methods sections 4.3 and 4.5 use Spatial predictability

Similarly for high-level predictability: high(er)-level feature predictability

Overall, I find simplifying some of these variations would improve readability for the reader and improve the manuscript.

I have a few minor comments:

Figure 1 seems a little crowded, (all) methods could be less described - I don’t have a clear suggestion.

Figure 3.b) example images are helpful and intuitive for the concepts of high vs. low level predictability, the authors might consider including more example images if there is room for it.

Typo on page 3 of manuscript: indepdent

Typo on page  5 of manuscript, Fig. 3 caption: unpredictabillity

Typo on  page 9 of manuscript: chacterised

Typo on  page 19 of manuscript: Figure S2 caption: feautre

Reviewer #3: This manuscript proposes to test predictive coding in data from mouse visual cortex. The authors use publicly available recordings from Allen Institute, including thousands of neurons from multiple visual cortical areas, while animals passively view static natural images flashed briefly on the screen. They use deep generative model based on the UNet architecture to quantify spatial predictability, namely how well a masked portion of the image can be reconstructed from the rest of the image (the surround or spatial context). They then use a linear regression approach to quantify the sensitivity of neurons to image predictability, namely the improvement of a linear fit to the firing rate across multiple images, when adding predictability as an independent variable in addition to low-level image features (e.g. contrast) and non-visual variables like running speed. The main finding is modulation by unpredictability, i.e. higher firing rate for unpredictable images, suppressed firing rate for predictable images. Another interesting finding is that the modulation is stronger when predictability is defined by deep feature layers (presumably more closely related to perception) than low-level features. Conversely, the base encoding model relies more on low-level features. This is interpreted as consistent with the idea that predictions are computed recurrently and via feedback, while feedforward processing encode low-level features.

Overall, the approach is interesting but somewhat underdeveloped, with some important choices not motivated; the theory is only outlined in words so some of the predictions (e.g. about layer specificity) are difficult to interpret in the data; and the main effects in the data are small to the point that it is unclear how meaningful they are. Some key references are missing and/or not sufficiently discussed. The manuscript is written clearly although some technical details could be added. Details are provided below.

1. Definition and quantification of predictability

This is the key metric, but not much is done to establish that it is a valid metric. It is not obvious that it is a rigorous perceptual predictability metric, and even less so that it is meaningful for visual cortex neurons. For the latter, one important control would be to apply it to gratings, bars, textures, noise stimuli etc. with which contextual effects have been studied in V1, and test if predictability so defined reproduces classical effects. If it does, is there something unique about this metric, or would any other intuitively meaningful metric of predictability/redundancy work as well? If it does not, how is one to interpret the results, i.e. that cortex is sensitive to predictability for natural images but not simpler parametric stimuli?

Relatedly, in classical studies with both parametric and natural stimuli, sensitivity to center-surround similarity is both stimulus-specific and neuron-specific (e.g. https://pubmed.ncbi.nlm.nih.gov/7477405/ ; https://pubmed.ncbi.nlm.nih.gov/12424293/). In contrast, this metric seems to be only image-specific, or feature layer specific in some analysis, except for the fact that it is computed by masking the image portion falling inside the RF of each neuron (if I got this right?). It is possible that a more granular estimation of predictability based on the per-neuron feature (not just the deep net layer) would reveal stronger sensitivity than the “perceptual” predictability for the entire image.

Another issue is that sensitivity to predictability is determined with a linear regression approach. Predict firing rate as linear combination of low-level features (e.g. contrast, orientation etc.) plus non-visual factors (e.g. running speed) and compare w/out predictability. This approach has become common (in other contexts), but it ignores that contextual effects in visual cortex are nonlinear and their nonlinearity has been formalized and characterized extensively (https://pubmed.ncbi.nlm.nih.gov/12424292/). So for instance the effects might differ if predictability were used to weigh the surround normalization term of E_norm in section 4.6.

2. Effect size and timing

The authors report modulation by unpredictability, i.e. higher firing rate for unpredictable images, suppressed firing rate for predictable images. This effect is smaller than modulation by low-level features, but significant. Nonetheless, the effect reported is quite small: Line 104 reports a DeltaR of 0.0036 over an R of approximately 0.2 (Fig. 2a,b), so a 5% effect. I understand it is statistically significant, but perhaps it is worth 1) understanding why it is stronger/weaker in some neurons than others (if indeed it is the case that the magnitude is variable across cases) and 2) speculate how such a small modulation could impact or be used by downstream computation.

The timing of the effect is also in line with the expectation that predictability requires more recurrent processing than low-level features, but again the effect is small, a 6 ms difference. How reliable and meaningful is that difference, given that the analysis of rate is based on 10 ms counting windows with a 30-ms smoothing?

The key reference is Uran et al 2022 where macaque V1 was shown to be sensitive to predictability. Line 55 states that in that paper sensitivity emerges 200-600ms after stimulus onset, whereas here images are presented for 250ms only and the analysis appears to be restricted to that window. The authors could comment on the reasoning why these presumably feedback/high-level effects are expected within the shorter window than in Uran.

One of the results I found most interesting is that the modulation by predictability is stronger for predictability defined by deep feature layers (presumably more closely related to perception) than low-level features. Conversely, the base encoding model relies more on low-level features. It would be interesting to know if these two effects also have different timecourse. For instance, does the timing of the effect of predictability vary when using superficial versus deep features to compute predictability?

3. The theory is underspecified

The Discussion on the laminar specificity of the effects expected by predictive coding raises two issues. First, given that the qualitative effects based on the abstract theory are not clear cut, it seems important to implement the theory and conduct simulations to provide more granular predictions. And second, the discussion closes by suggesting that more granular data analysis would be required too, but does not give any hint of what those experiments and analysis would entail. Overall, the paper would be much stronger if it were driven by an actual implementation of the theory, to better understand what aspects of predictive coding are distinctively supported by the data.

4. Additional comments and suggestions

The author could comment on the findings by Tolias group that suggest V1 selectivity is more complex, i.e. tuned to higher level features, than classically thought. This seems somewhat relevant to the dissociation reported here between sensitivity to low-level features versus predictability. https://pubmed.ncbi.nlm.nih.gov/31686023/

Surround modulation has been extensively linked to predictive coding. It seems appropriate to discuss the relation to that work. E.g. https://pubmed.ncbi.nlm.nih.gov/21315102/

Methods 4.4: the loss function includes multiple terms, with specific choices of their relative weights. It would be helpful to write down the key equations and explain them, so the reader doesn’t need to hunt them down in the referred papers.

Methods 4.5: what is the rationale for using Alex Net for the predictability analysis while the model uses VGG? How do we know that does not impact the results?

Fig 2 caption says ‘0-30 ms’ but methods says 20 ms?

Line 331, sentence starting with “Moreover” is repeated twice.

Line 336, define CCF.

Line 357 ‘receptor-field’ is it ‘receptive-field’ instead?

Line 380 ‘L2’ not properly formatted

Line 426 ‘this set comprehensive set’ remove the first ‘set’

Line 472 missing full stop before ‘Each’

**Have the authors made all data and (if applicable) computational code underlying the findings in their manuscript fully available?**

Reviewer #1: Yes

Reviewer #2: Yes

Reviewer #3: Yes

PLOS authors have the option to publish the peer review history of their article (what does this mean?). If published, this will include your full peer review and any attached files.

Reviewer #1: **Yes:** Binxu Wang

Reviewer #2: **Yes:** Cem Uran

Reviewer #3: No

**Figure resubmission:**
---

## [Decision Letter · Decision Letter 1]

26 Dec 2025

Dear Dr. Heilbron,

We are pleased to inform you that your manuscript 'Higher-level spatial prediction in natural vision across mouse visual cortex' has been provisionally accepted for publication in PLOS Computational Biology.

Best regards,

Xue-Xin Wei

Academic Editor

PLOS Computational Biology

Marieke van Vugt

Section Editor

PLOS Computational Biology

Reviewer's Responses to Questions

**Comments to the Authors:**

Reviewer #1: Thanks for the revision, the reviewer is satisfied and recommend acceptance! Congratulations.

There is one minor clarification question about R2 plot Figure R4 (details below)

Below, we briefly reply to each of the response to the major concern points.

### *Opposing trends in feature predictivity vs. unpredictability modulation*

Thanks for the new synthetic control. It makes the observed anti correlation stronger.

### *Statistical robustness of the depth–predictivity relationship (Figure 4A)*

Seems to me that the Figure 4B still shows quite fuzzy effect.

From the time resolved analysis, the effect seems to be consistent across time windows.

### *Delta R2*

We appreciate the authors plotting the figures again using R2.

Regarding DeltaR2, indeed, the effect size is relatively small, but consistent and statistically robust.

I agree that, the R2 will be a more stringent measure of the model goodness of fit. When there is scaling or baseline shift between training and test sets, then we will observe that R2 is smaller than square of r (r2). The latter is equivalent to re-fit a shift and scaling for the firing rate on the test set.

One additional question is that, I’m still not quite sure **why the baseline period have a negative R2, in Figure R4**. Could it be that the authors didn’t add a bias term to the predictive model? or it’s due to baseline firing rate change between train and test split? I feel if the train and test are randomly selected from the shuffled trials, the baseline firing rate should match relatively well, then the R2 should be 0 at least. But maybe my intuition is wrong

**In general I feel it will be helpful to include figure R3 R4 (or other plot of R2) in supplementary to give audience a reference number for the explained variance change.**

### *Interpretation of the Split-Half Analysis Results*

Thanks for the clarification, now it’s very clear to me.

Reviewer #2: I have reviewed the revised manuscript and appreciate the authors' detailed responses. I have no further comments.

Reviewer #3: The authors have address all my comments extensively. I appreciate the additional analysis they conducted, I believe the addition of Fig. R8,9 as supplements is appropriate and adds value to the paper. Most other points required clarification, which have been provided. The only outstanding issue is that relating the findings to the theory(ies) of predictive coding is very difficult short of a formulation of the theory that actually makes predictions for this data/analysis; but I agree with the authors that that's beyond the scope of this paper. I commend the authors for doing an excellent job on the rebuttal and congratulate them on a thorough and interesting paper.

**Have the authors made all data and (if applicable) computational code underlying the findings in their manuscript fully available?**

Reviewer #1: Yes

Reviewer #2: Yes

Reviewer #3: None

PLOS authors have the option to publish the peer review history of their article (what does this mean?). If published, this will include your full peer review and any attached files.

Reviewer #1: No

Reviewer #2: **Yes:** Cem Uran

Reviewer #3: No

---

## [Editor Report · Acceptance letter]

PCOMPBIOL-D-25-00957R1

Higher-level spatial prediction in natural vision across mouse visual cortex

Dear Dr Heilbron,

I am pleased to inform you that your manuscript has been formally accepted for publication in PLOS Computational Biology. Your manuscript is now with our production department and you will be notified of the publication date in due course.

With kind regards,

Anita Estes
